# Metformin: Expanding the Scope of Application—Starting Earlier than Yesterday, Canceling Later

**DOI:** 10.3390/ijms23042363

**Published:** 2022-02-21

**Authors:** Yulia A. Kononova, Nikolai P. Likhonosov, Alina Yu. Babenko

**Affiliations:** National Almazov Medical Research Centre, Institute of Molecular Biology and Genetics, 197341 Saint-Petersburg, Russia; ncmu@almazovcentre.ru (Y.A.K.); likhonosov.pn@gmail.com (N.P.L.)

**Keywords:** diabetes mellitus, obesity, metformin, molecular mechanisms, atherosclerosis, cardio-vascular diseases, heart failure, chronic kidney disease, lactate

## Abstract

Today the area of application of metformin is expanding, and a wealth of data point to its benefits in people without carbohydrate metabolism disorders. Already in the population of people leading an unhealthy lifestyle, before the formation of obesity and prediabetes metformin smooths out the adverse effects of a high-fat diet. Being prescribed at this stage, metformin will probably be able to, if not prevent, then significantly reduce the progression of all subsequent metabolic changes. To a large extent, this review will discuss the proofs of the evidence for this. Another recent important change is a removal of a number of restrictions on its use in patients with heart failure, acute coronary syndrome and chronic kidney disease. We will discuss the reasons for these changes and present a new perspective on the role of increasing lactate in metformin therapy.

## 1. Introduction

More and more data demonstrate that the start of a metabolic disorders cascade leading to both diabetes mellitus (DM) and associated cardiovascular (CV) problems occurs with a hypercaloric diet high in fat and carbohydrates with a high glycemic index. As current studies have shown, such nutritional characteristics are associated with an increased risk of cardiovascular diseases (CVDs) even before the development of obesity and diabetes [1,2,3]. Normally, excess nutrients are deposited in subcutaneous adipose tissue (SAT); however, a metabolic health disorder is associated with a shift in the deposition of fats in the visceral depot and with deposition because of hyperplasia of adipocytes to their hypertrophy, an increase in volume [4,5].

Mechanisms of reprogramming the location and nature of the deposition of excess nutrients are being actively studied. The way of an expansion and remodeling of white adipose tissue (WAT) directly affects the risk of insulin resistance (IR) and metabolic syndrome development in obesity. Predominant WAT accumulation in the visceral depot is associated with an increased risk of IR, whereas subcutaneous WAT accumulation is protective for metabolic health. The processes of nutrient deposition can also differ—occurring either by hyperplasia of adipocytes or by hypertrophy [6]. Pathological remodeling of WAT is characterized by adipocyte hypertrophy accompanied by chronic inflammation and fibrosis and is associated with IR. A “healthy” variant of nutrient accumulation suggests WAT hyperplasia, where adipocytes have a significantly smaller volume, and the severity of inflammation and fibrosis is minimal [6]. This “metabolically healthy” variant of obesity is characterized by normal insulin sensitivity and the absence of metabolic disorders, including type 2 DM.

The role of nutrient exposure through the microbiome composition and functioning in the transition from metabolically healthy to metabolically unhealthy obesity.

In recent years, several mechanisms linking the gut microbiota with the development of obesity and associated metabolic disorders (IR, type 2 diabetes) have been identified. There is accumulating evidence of a causal relationship between the imbalance in normal populations of microorganisms in the human gastrointestinal tract (GIT) and the development of obesity, IR, type 2 DM, and CVD [7,8]. Bacteria of the phylotypes Bacteroidetes and Firmicutes predominate in the microbiota of the digestive tract in healthy adults [9,10]. Glyco- and lipotoxic effects of nutrients rich in animal fats and rapid carbohydrates lead to an imbalance in various subtypes of bacteria that form the human GIT microbiota [11]. With the progress of obesity, a relative decrease in Bacteroidetes in relation to Firmicutes occurs, the contribution of Actinobacteria changes, and the numbers of butyrate and lactate-producing bacteria decrease [12]. Type 2 DM is characterized by a greater imbalance in the composition of the microbiome (Table 1) [13,14,15,16,17,18].

The gut microbiota interact with the host organism through recognition receptors—Toll-like receptors (TLRs). Obese people demonstrated a significantly greater increase in plasma lipopolysaccharide (LPS) levels after a high-fat meal compared to non-obese people [42]. Translocation of LPSs across the intestinal mucosa is a characteristic of obesity, type 2 DM, and related disorders. Eating a high-fat diet dose-dependently elevates the production of LPSs by Gram-negative bacteria in the intestine and facilitates their entry from the intestine into the blood. The leakage of LPSs into the blood activates the nonspecific inflammation affecting the metabolism of the liver, adipose, and muscle tissue. In addition, these endotoxins can alter the activity of the small intestine nervous system and the gut–brain axis through the vagal nerve, affecting the appetite regulation. This concept of metabolic endotoxemia (an increase in plasma LPSs) is considered one of the triggers leading to the development of meta-inflammation and IR [48]. Eating a high-fat diet increases the production of Gram-negative bacteria in the intestine and the uptake of LPSs from the intestine into the bloodstream. LPS production correlates with the content of lipids in food. Both lipids and LPSs produced by Gram-negative bacteria in the microbiota bind scavenger receptor class B type I (SR-BI), which enhances the incorporation of LPSs into chylomicrons. These complexes are transported through the lymph into the bloodstream, where LPSs are transferred to other lipoproteins, mainly high-density lipoproteins (HDLs) by translocases. LPSs bound by SR-BI enhance lipoprotein transcytosis across the endothelial barrier and endocytosis in adipocytes [10]. Adipocytes, which absorb the most LPS-rich lipoproteins, become large. In addition, macrophages in adipose tissue internalize LPS lipoproteins, which can change their phenotype from M2 to M1. Large adipocytes are more metabolically active and uptake more LPSs than small adipocytes. LPSs within adipocytes activate caspase-4/5/11, which can induce a highly inflammatory type of programmed cell death (pyroptosis). Adipocyte death occurs when their size grows to such an extent that the intracellular concentration of LPSs initiates pyroptosis. Thus, the hypertrophy, inflammation, and death of adipocytes are stimulated through the mechanisms that cause the excessive production of LPSs by the intestinal microbiota [46]. This means that the diet-induced disturbances in the composition of the intestinal microbiome contribute to the development of a hypertrophic variant of obesity and the occurrence of chronic inflammation. As a result, developing chronic inflammation can lead to metabolic dysregulation in many organs (the intestine, the adipose tissue, the muscles, the liver, and the brain), in particular, through the modulation of the innate and adaptive immune system [59].

Another mechanism for the induction of not only adipose tissue inflammation, but also inflammation of other organs is a change in the effects of specific metabolites produced by intestinal bacteria, primarily short-chain fatty acids (SCFAs) (lactate, butyrate, propionate, acetate, and succinate). These metabolites can affect the local and the systemic immune response and modulate metabolic homeostasis [60]. Enteroendocrine cells (EECs) producing glucagon-like peptide-1 (GLP-1), peptide YY (PYY), and glucagon-like peptide-2 and the endocannabinoid system have also been demonstrated to control the intestinal permeability and to be involved in metabolic endotoxemia and altered signal transmission from the GIT to the central nervous system. The digestive tract interacts with the central nervous system, transmitting the information about the nutritional status through a variety of mechanisms, including EECs, the vagus nerve system, and the enteric nervous system. SCFAs can modulate the secretion of these hormones and neurotransmitters [61,62]. The vagus nerve is involved in appetite regulation and gastrointestinal motility and has anti-inflammatory properties [63]. Mechanoreceptors of the GIT, hormones (ghrelin, PYY, cholecystokinin, GLP-1) secreted by the EECs and microbial metabolites (SCFAs) are involved in its activation (ghrelin and PYY inhibit it; cholecystokinin and GLP-1 activate it) [64]. In addition, SCFAs themselves are the signaling molecules that bind to G-protein-coupled receptors (GPRs) GPR43 and GPR41 and modulate satiety and may increase energy expenditure [65]. They also play a role in central nervous system (CNS) inflammation [66]. These changes, in turn, contribute to the impairment of incretin production. Most likely the signaling alterations from the GIT, through both a change in the production of microbiota metabolites (SCFAs) and the modification of the hormone production in the GIT, lead to the reprogramming of the deposition pathway in the SAT from hyperplasia (which shows metabolically healthy obesity) to hypertrophy of adipocytes and increased fat accumulation in visceral adipose tissue (VAT) with the formation of adipose tissue ectopia and changes in the functions of organ adipocytes. Glucose-dependent insulinotropic peptide (GIP), which is responsible for the deposition of lipids in the SAT under the normal conditions, is the likely key factor of the reprogramming [67]. The role of GIP in the regulation of energy accumulation in adipose tissue and the development of metabolic disorders was confirmed by the fact that GIP affects all the key tissues that are important for the control of glucose and lipid homeostasis, stimulates insulin biosynthesis and secretion, and increases the viability of islet cells. GIP regulates lipid metabolism (modulating lipolysis and lipogenesis depending on insulin level) directly through its receptor on adipocytes. In the fasting state GIP stimulates glucagon secretion and lipolysis in the SAT, and in the postprandial state (an increase in glucose and insulin levels) it inhibits glucagon secretion, stimulates insulin secretion and adipogenesis in the SAT, and increases the intake of triglycerides (TGs) in the SAT. Normally, GIP is responsible for the deposition of excess energy in the SAT; consequently, an impairment of GIP signaling can contribute to changing the fat accumulation from the SAT to the VAT.

Thus, a change in the composition and functioning of the microbiome through the mechanisms, including a change in the production of active metabolites—in particular, SCFAs—leads to a change in intestinal permeability, the production of incretins and other hormones of the GIT. The impairment of their signaling to a large extent contributes to a change in fat accumulation from the SAT to the VAT, from hyperplasia to hypertrophy, the development of IR in adipose tissue, chronic inflammation, and OS. In recent decades, five stages have been conventionally distinguished in the chronology of type 2 DM development. The earliest stages, which are characterized by the absence of carbohydrate metabolism disorders, are the 1st stage (latent) and the 2nd stage (transient) [68]. They are characterized by hyperinsulinemia, IR, and activation of the renin–angiotensin–aldosterone system both systemically and in the pancreas islets. Pathophysiologically, this comprises the islet redox stress (oxidative and nitrosative stress), free-radical polymerization of islet amyloid polypeptide monomers, β-cell endoplasmic reticulum (ER) stress, and unfolded protein response (UPR) stress. At the 2nd stage, the processes of amyloidosis and fibrosis of β-cells join [69]. They become the basis for the development of further carbohydrate metabolism disorders. This means that the prevention of IR development should be considered the earliest option for the prevention of all further pathological changes.

## 2. Metformin Role in Prevention and Correction of the Changes Associated with the Fat-Rich Food and High Glycemic Index Products, Effects on the Structure of the Microbiota

In recent years, the focus on the key effector organs has moved from the liver to the GIT. The most significant argument in favor of the GIT is the fact that intravenous injection of metformin does not have glucose-lowering effects [70,71], and its concentration in the intestinal mucosa is 30–300 times higher than in plasma [72]. The fact that metformin positively affects the GIT, even if the pathological stereotype of the nutrition persists, is extremely important. Today, one way of realizing the effects of metformin is through the strengthening of incretin actions (Table 1) [73]. The mechanisms of metformin interaction with the intestinal microbiome include not only the regulation of glucose metabolism, but also an elevation in the production of SCFAs, an increase in the intestinal permeability for LPSs, the modulation of the immune response, and interaction with bile acids [73] (Table 1).

Bile acids (cholic acid and chenodeoxycholic acid) are another significant recipient of metformin’s effects. Bile acids are synthesized from cholesterol in the liver and are secreted into the intestine, where they are converted into secondary bile acids such as deoxycholic acid and lithocholic acid with the participation of the enzymes and the intestinal microbiota (Table 1). They play an essential role in the metabolism of glucose and lipids [74] via a number of metabolic pathways, binding to several intracellular nuclear receptors, including the farnesoid X receptor (FXR), pregnane X receptor (PXR), and G-protein-coupled receptors (GPRs) [75]. The results of a meta-analysis of metagenomic data are a significant argument supporting the importance of metformin’s effects on the microbiota. This meta-analysis showed that the gut microbiome was less abundant in patients with type 2 DM without metformin treatment, and its diversity recovered almost to the level of healthy people in the metformin-treated control group [76].

Thus, the effects of metformin on the GIT microbiota and the inhibition of the harmful effects of disordered dietary stereotypes (fat-rich and high glycemic index foods), both on the composition of the microbiota and on the subsequent systemic effects, are the earliest in the prevention of the development of metabolic disorders leading to obesity, IR, type 2 DM, and its CV complications. Experimentally, treatment with metformin for 14 weeks in mice was shown to significantly prevent high-fat diet-induced obesity and associated inflammatory response by increasing the expression of fibroblast growth factor 21 (FGF-21), which is a key metabolic hormone improving lipolysis in brown adipose tissue (BAT) to prevent fat accumulation [77]. In addition, metformin can prevent obesity in mice by increasing the metabolic activity of BAT [78].

The effects of metformin on the intestinal permeability, LPS entry into the bloodstream, modulation of the gut–CNS axis, and production of SCFAs and bile acids can prevent (and if not completely, then significantly reduce) the severity of pathological reprogramming of fat storage from hyperplastic adipocytes of the SAT into hypertrophic adipocytes of the VAT, the development of meta-inflammation, and IR.

The recipients of metformin’s effects in patients with prediabetes and diabetes have been described in numerous reviews and are not a new topic. The mechanisms of metformin’s effect on the cardiovascular system deserve more attention. Although they are not a new topic, but their structuring and refinement considering the new data, will undoubtedly be useful.

DM has long been designated as the equivalent of cardiovascular pathology, and indeed it sharply increases the risk of cardiovascular death from two complexes of cardiovascular problems: atherosclerotic CVD and cardiorenal diseases (heart failure (HF) and chronic kidney disease (CKD)). The risk of ischemic heart disease is increased 3.77 (1.74; 8.17) times by diabetes [79]; half of type 2 DM patients have CKD [80], and 15% have chronic HF [81], which is 2.5 times more than in people without type 2 DM [82].

These two sets of problems are associated with the different mechanisms and develop at different stages of the cardiometabolic continuum. There is a growing body of evidence demonstrating that obesity plays a leading role among the factors associated with the early development of HF and CKD. Thus, the ARIC study analyzed the relationship between body mass index (BMI) and main CVD in linear models adjusted for other risk factors (age, sex, smoking, alcohol, physical activity, the presence of DM, the level of blood pressure, lipids, and glomerular filtration rate (GFR)). The most pronounced association with BMI was shown for the risk of HF [83,84]. Without a doubt, HF can be an outcome of atherosclerotic CVD, but this phenotype is formed much later and is associated with a complicated course of ischemic heart disease (myocardial infarction (MI)), a longer duration of DM, and insulin therapy [85,86].

The mechanisms that are the cornerstone of these processes are actively being studied. The violation of autophagy processes is among them. Its activation again may occur in the 1st–2nd stages of the formation of DM, before the development of dysglycemia [87] (Table 2).

The state of excess nutrients is characterized by the suppression of autophagy, the process responsible for the destruction of damaged organelles. Glucose and lipid metabolites such as diacylglycerol (DAG) inhibit the formation of autophagic vacuoles and their fusion with lysosomes for lysis. The suppression of autophagy leads to the accumulation of damaged organelles, which are the main substrate for oxidative stress (OS) and ER stress [143].

The general mechanisms of the formation of myocardial dysfunction and HF in IR and type 2 DM are based on hyperinsulinemia. Today hyperinsulinemia is considered to be the pivotal trigger for the development of HF in obesity and type 2 DM. The key and universal mechanism for any HF phenotype is an increase in the activity of sodium–hydrogen exchangers of types 1 and 3 (NHE1 and NHE3) [144] (Table 2). The NHE1 isoform is expressed in all organs, but it predominates in the heart, where it regulates cell volume and pH, and NHE3 isoform expression is limited to the apical surface of the renal tubules and epithelial cells of the GIT, where it provides sodium reabsorption. Hyperinsulinemia increases NHE1 and NHE3 activity, and thus, type 2 DM may contribute to the development of both HF with reduced ejection fraction (HFrEF) and HF with preserved ejection fraction (HFpEF) [89].

Hyperinsulinemic, metabolically unhealthy obesity is characterized by the accumulation of not only visceral but also ectopic fat, which is involved in the induction of myocardial inflammation, microcirculation dysfunction, and fibrosis [145,146] (Table 2).

In healthy people, epicardial adipose tissue is BAT; however, under the conditions of excess fatty acids, lipotoxicity, and imbalance of adipokines (hyperleptinemia and leptin resistance, excess tumor necrosis factor α, interleukin-6 and resistin, adiponectin deficiency), a rise in cell volume and the redifferentiation of epicardial adipose tissue cells occurs, wherein it acquires the properties of WAT. Considering the conjoint microcirculatory system of the epicardium and myocardium, when metabolic functions and the secretory profile of epicardial adipose tissue adipocytes change, the myocardium is exposed to the produced pro-inflammatory cytokines being involved in the processes of inflammation and fibrosis [147]. The significance of these relationships is supported by the fact that the volume of epicardial adipose tissue is closely related to the severity of coronary capillary dysfunction, myocardial fibrosis, and LV hypertrophy [147]. Hyperleptinemia and leptin resistance, which are formed, such as hyperinsulinemia/IR, at the metabolically unhealthy obesity stage, contribute significantly to the described changes, exacerbating the disorders induced by hyperinsulinemia [91]. The severity of hyperleptinemia correlates with the volume of epicardial adipose tissue, promotes sodium retention through the stimulation of aldosterone secretion, enhances myocardial and vascular fibrosis, and, accordingly, relaxation disorders, and negatively affects calcium metabolism in the myocardium [94,147].

Hyperuricemia is among the disorders involved in the early pathogenesis of obesity and metabolic syndrome. Hyperuricemia has been identified not only as a risk factor for gout, but also as a contributing factor to impaired glucose metabolism, dyslipidemia, and hypertension. High uric acid levels are strongly associated with cardiovascular disease, including coronary artery disease and HF [148,149].

Hyperuricemia has been demonstrated to be an inducer of oxidative stress in many cells [150], including hepatocytes and pancreatic endocrinocytes [112] and kidneys cells [151] (Table 2).

Hyperuricemia is characterized by an increase in inflammation markers such as C-reactive protein (CRP), fibrosis markers such as galectin-3 (Gal-3) and carboxy-terminal telopeptide of type I collagen (CITP). In general, a high level of these markers as well as procollagen-3, according to a number of studies, is specific for patients with obesity and type 2 DM and is a predictor of the development of CV disorders (HF, atrial fibrillation, progression of atherogenesis).

## 3. Key Mechanisms for the Development of CKD and HFpEF

CKD and HFpEF form a cardiorenal complex that begins long before the development of DM and is associated with IR and hyperinsulinemia. This has been proven by numerous experimental and clinical data [152,153,154,155]. In fact, hyperinsulinemic obesity and type 2 DM are states of chronic excess energy, in which the Akt/mTORC1 pathway is activated and the suppression of sirtuin 1 (SIRT1), which is activated by caloric restriction and the suppression of its descending executive elements—peroxisome proliferator-activated receptor gamma coactivator 1a (PGC-1a), fibroblast growth factor 21 (FGF-21), and adenosine monophosphate-activated protein kinase (AMPK)—occurs [156].

During embryogenesis, the activation of Akt/mTORC1 promotes the growth and development of the heart; however, in adults, the activation of this pathway leads to disruption of the autophagy process, and hyperactivity from hyperinsulinemia causes HF, while suppression of Akt signaling reduces the severity of hypertrophy and fibrosis of the myocardium and delays the development of HF [156]. AMPK in the myocardium counteracts Akt/mTORC1. As a result, type 2 DM is characterized by the decreased activation of SIRT1/PGC-1a/FGF21 and AMPK, as well as by autophagy suppression. These disorders precisely distinguish hyperinsulinemic states, since insulin directly suppresses autophagy through the inhibition of SIRT1 and activates Akt/mTORC1 signaling [157,158].

The mechanisms of lipotoxicity include an increase in the concentration and oxidation of fatty acids (FAs) in the myocardium because of the growth in their uptake. With the activity of peroxisome proliferator-activated receptor-α (PPAR-α), the accumulation of TG and DAG in the myocardium increases. The uptake and the oxidation of glucose, on the contrary, are reduced owing to a decrease in the activity of the glucose transport protein (GTP)-4 (Table 2). An important role in the development of contractile disorders is played by a decreased Ca2 + turnover in the cardiomyocytes [159]. Increased activity of protein kinase C (PKC) reduces the availability of nitric oxide and causes an increase in endothelin-1 production and *connective tissue growth factor* expression. An elevated AGE formation with a violation of the AGE/receptor for advanced *glycation* end products (RAGE) ratio is accompanied by a reduced production of transforming growth factor β (TGF-β) and of ROS. The activation of the renin–angiotensin–aldosterone system accompanies these changes.

Thus, the predominant mechanism for the progression of HFpEF in DM, primarily against the background of obesity and metabolic syndrome, is characterized by the induction and the expansion of inflammation (systemic and in epicardial adipose tissue). Inflammation causes mediated by pro-inflammatory cytokines microvascular dysfunction, oxidative stress, and myocardial fibrosis. Hyperuricemia and changes in microbiome composition make important contributions to the induction of inflammation. We have already considered the change in the characteristics of the microbiome as the first step in a cascade of inflammatory changes and changes in the characteristics of adipose tissue (the transition of hyperplasia to hypertrophy), which form its metabolically unhealthy phenotype. As described in detail above, hyperinsulinemia and hyperuricemia, typical for obesity and metabolic syndrome chronic inflammation, through transforming growth factor-β (TGF-β), AGE and ROS-dependent pathways, induce the development of myocardial, vascular, and renal fibrosis. These changes are key to the progression of HFpEF and CKD.

## 4. Pathogenetic Features of Atherosclerosis Development in IR and Type 2 DM

Atherosclerosis is a major diabetes complication with several causative factors. Vascular endothelial dysfunction (ED) characterizes both DM and atherosclerosis. Hyperglycemia, in addition to lipotoxicity and hyperinsulinemia, plays a significant role in the activation of ED, inflammation of the vascular wall, and atherogenesis. The crucial mechanisms of ED in vascular injury and atherogenesis are presented in Table 2 [113,114,115,118].

At high glucose levels, the proatherogenic pathway is activated to a greater extent (Table 2) [119,120]. Vascular inflammation is a key change leading to atherosclerosis, and DM characterized by an increase in circulating markers of inflammation and expression of monocytic genes of pro-inflammatory mediators worsens it [122]. Monocytes migrating into the endothelium give rise to the conversion of foam cells with an accumulation of lipids and the formation of atherosclerotic plaques. The resulting atherosclerosis underlies a number of atherosclerotic CVDs.

Hyperglycemia alters the expression of a number of miRNAs that affect both vascular dysfunction and inflammatory conditions [123]. They are miR-126, miR-21, and miR-146a-5p. These changes are associated with changes in NF-κB activity. Activation of miR-21, according to some authors, may be a critical event linking diabetes and the development of atherosclerosis (Table 2). miR-146a-5p has been identified as a negative regulator of NF-κB [128,160] (Table 2).

Finally, vascular calcification is among the key features of vascular lesion in DM, which is a severe, irreversible change and typical for people with atherosclerosis associated with diabetes and CKD (Table 2). An imbalance in the AGE/RAGE axis in hyperglycemia promotes vascular calcification, exacerbating preexisting atherosclerotic damage with more severe changes. Previous studies have shown the association between plaque formation in the carotid arteries in DM, increased glycosylation of N-acetylglucosamine (O-GlcNac), and stimulation of the osteogenesis regulator Runt-related transcription factor 2 (Runx2) [135]. Atherosclerosis is characterized by intimal calcification, but DM can induce medial calcification even in the absence of atherosclerosis (Table 2). In addition, miR-126, miR-21, and miR-146a are common in diseases that play a regulatory role in vascular calcification of the media and the transdifferentiation of vascular smooth muscle cells into osteogenic-like cells [114]. The combination of intimal and medial calcification is observed mainly in peripheral medium caliber arteries and associated with the progression of atherosclerosis and serious complications from cerebrovascular vessels (dementia, stroke), renal vessels (chronic renal failure), and lower extremities (critical ischemia).

## 5. Metformin’s Effects on the Inhibition of the Mechanisms of Cardiorenal Continuum Formation—HFpEF and CKD

Above, we identified the main stages in the formation of HFpEF and CKD through a pathological lifestyle and an imbalance in the composition of the microbiota into metabolically unhealthy obesity, hyperinsulinemia, and IR, with the progression of chronic inflammation, oxidative stress, and the development of fibrosis. The mechanisms of metformin’s influence on these processes were also largely, but not completely, discussed above. As discussed in the recent review, metformin blocks the development and the progression of both HFpEF and HFrEF through a variety of complex pleiotropic effects, in particular, by its direct effects on myocardial structure and function, allowing the maintenance of normal LV morphology and functional activity [161]. These mechanisms are described in detail in this review, so we will not dwell on them here.

Hyperuricemia has been identified as an important factor in oxidative stress and activation of fibrogenesis in the myocardium and kidneys. Metformin in experimental rat models decreased serum uric acid levels and, through cyclic adenosine monophosphate (AMP)-dependent protein kinase, reduced the negative effects of hyperuricemia [108,162]. Metformin can reduce the severity of hyperuricemia by activating AMPK and its phosphorylation and protecting against IR caused by hyperuricemia in cardiomyocytes, skeletal muscles [111], and associated pathological processes, including increased fibrogenesis (elevated level of Gal-3, procollagen types 1 and 3).

Metformin, as a powerful AMPK activator, is a promising drug for reducing or reversing the fibrosis. Numerous studies examining the mechanisms of metformin on fibrosis have shown that it mainly exerts an anti-fibrotic effect, affecting the signaling pathway of TGF-β, cellular metabolism, and oxidative stress, including that induced by hyperuricemia. Metformin has a direct anti-fibrotic effect (Table 2), inhibiting the production of TGF-β1 [88,95,96,97,99,100,101,103].

In a hybrid study involving in vitro experiments, animal models, and clinical evaluation of patients, its production was shown to be higher in the VAT than in the SAT and in circulation. Gal-3 showed a positive BMI-dependent correlation with leptin, resistin, interleukin-6, and age [163]. In patients with type 2 DM, Gal-3 was elevated in serum. It positively correlated with the level of C-reactive protein. Metformin treatment was associated with lower levels of Gal-3 in the systemic circulation in patients with type 2 DM (Table 2). The ability of this drug to reduce the severity of oxidative stress and the formation of AGEs [164], which, in turn, induce the expression of Gal-3, is considered as a possible mechanism for reducing the level of Gal-3 during metformin therapy. The contribution of AMPK activity modulation is also being considered. Oxidative stress is a central mechanism involved in fibrotic progression. AMPK is the oxidative stress suppressor and is critical for regulating ROS production. Metformin, as an AMPK activator, reduces ROS production and suppresses oxidative stress [99,103,107,165,166,167].

The cardioprotective effects of metformin have been largely determined by its ability to decrease the severity of cardiac fibrosis caused by pressure overload, MI, and many other causes [95,107,168,169,170]. Similarly, experimental renal fibrosis associated with high-fat diets [171], unilateral ureteral obstruction [172], high-dose folate intake [96], adenine [173], and cyclosporin A [174] was significantly reduced with metformin. Accumulated data have shown that metformin has great potential in reducing the progression of fibrosis in various organs, including the VAT [174,175], SAT [176], uterus and ovaries [177], lungs [88,170], and liver [178]. At the same time, metformin decreased not only the severity of fibrosis induced by hyperinsulinemia, oxidative stress, and hyperglycemia in metabolic syndrome and type 2 DM, but also fibrosis of a different genesis—ischemic, toxic, and drug induced.

Interestingly, metformin administration did not alter AMPK and p38 MAPK activity or collagen levels in TGF-β1 or cardiac fibroblasts with high glucose content [179]. This may support the idea that the anti-fibrotic effects of metformin are also most pronounced at the stages of high-fat nutrition, hyperinsulinemia, and IR prior to the formation of severe hyperglycemia. At the same time, metformin can reduce the hyperglycemia-induced inhibition of the B1/AMPK/Akt pathway, activate GSK3-β, and prevent diabetes-induced cardiomyopathy [102], suggesting that, while some mechanisms of the protective effects of metformin are weakened under hyperglycemic conditions, others are characterized by high functionality.

Nearly all the mechanisms underlying the development of HFpEF and diabetic kidney injury are blocked by metformin. In general, they reflect the universal processes of diabetic damage under the conditions of hyperinsulinemia and hyperglycemia: changes in the signaling of the Akt–AMPK–mTOR axis, induction of ER stress and epithelial-to-mesenchymal transition (EMT), inhibition of autophagy processes, generation and accumulation of ROS and AGE; under the influence of chronic hypoxia, an increase in hypoxia-inducible factor (HIF) activity, lipotoxicity, including inhibition of CREBP1 and FAS occurs. ER stress under hyperglycemic conditions is caused by protein accumulation, ROS generation, and mTOR activation in both cardiomyocytes and renal epithelial cells. AMPK activation by metformin protects the myocardium, renal epithelial cells, and other tissues from ER stress by inhibiting the long-term UPR, ROS generation, and mTOR activation. In obesity and type 2 DM, ER stress, which develops under the influence of hypoxia and/or glucose deficiency in the cell, plays an essential role in the activation of the expression of glucose-regulated protein 78 (GRP78). Recently, data on the ability of metformin to influence its activity have been obtained [180].

Moreover, the modulation of GRP78 activity by metformin turned out to play a significant role in its antitumor activity [181] and in improving the prognosis in COVID-19. Modulation of GRP78 activity may contribute to the implementation of many metformin action mechanisms. Normally, GRP78 is localized in the ER, where it plays an important role in the folding and assembly of proteins and, conversely, in the “re-folding” of misfolded/unfolded proteins or their degradation [181].

Under pathological conditions, such as during hypoxia, glucose starvation, in cells infected with fungi and viruses, and in tumor cells, this protein is overexpressed on the cell membrane and can be detected in the circulation in a soluble form. Overexpression of GRP78 on the cell surface facilitates the entry of pathogens (bacterial, fungal, and viral) into the cells and increases the aggressiveness of cancers. The pronounced overexpression of GRP78 in adipose tissue and pancreatic cells in visceral obesity, which is a consequence of hypoxia and intracellular glucose deficiency as a result of IR, is most likely the primary target of metformin in realizing its effects on adipose tissue health, improving the prognosis in COVID-19, and improving the response to treatment of many tumors. In fact, improving cell viability under hypoxic conditions can be identified as the key effect of metformin. Hypoxia and an increase in the level of angiotensin II under its influence in diabetic nephropathy cause the elevation in intrarenal and systemic pressure and the accumulation of collagen. These processes are accompanied by hypoxia-induced rise in HIF1a. The ability of metformin to reduce the severity of these processes has been confirmed by the effect of metformin on the HIF1a level in diabetic nephropathy, which is realized through the inhibition of the mitochondrial respiratory complex I and the oxygen redistribution in the cells. Metformin reduces oxygen demand and consumption and lowers ATP levels, which in turn promotes proteasome degradation of HIF1a and prevents hypoxia-induced damage to renal epithelial cells. Metformin may also protect podocytes in diabetic nephropathy [182]. Metformin, blocking the AGEs–AGER–ROS axis, exhibits its antioxidant effect and reduces the risk of cellular damage, which has been found to contribute to non-diabetic kidney disease, thus reducing the risk of kidney stones formation. Metformin decreases the severity of inflammation, the penetration of immunocytes into the renal epithelium, and modulates their functions (including in acute renal injury), activates the autophagy process in renal diseases, and reduces apoptosis in an AMPK-dependent way, through mechanisms similar to those described above for HFpEF. Finally, metformin reduces lipotoxicity by inhibiting the cellular damage processes even at the stage of obesity, improving lipid metabolism and protecting cardiomyocytes and mesangial kidney cells from apoptosis caused by lipotoxicity. As already noted, metformin resolves or inhibits pathological processes in non-diabetic renal damage as well. In addition to the abovementioned effect on urolithiasis development, metformin inhibits the progression of autosomal dominant polycystic kidney disease, interfering with cell proliferation and inhibiting cystic fibrosis transmembrane conductance regulator (CFTR) and mTOR signals through AMPK. The ability of metformin to protect the kidney from the toxic effects of nephrotoxic drugs deserves special attention. It protects the renal tubules from damage by regulating oxidative stress and repairing biochemical changes.

An important metformin effect on inhibiting the progression of virtually any renal pathology is its ability to protect against the development of renal fibrosis. Metformin slows down the renal hypoxia-induced fibrosis by inhibiting the stabilization of HIF1α, decreasing renal oxygen consumption, reducing the TGF-β1 level, and blocking its binding to the receptor.

## 6. Clinical Findings for the Cardioprotective Effects of Metformin on the Cardiorenal Continuum (HFpEF and CKD)

Currently, the effects of metformin therapy have been evaluated in different HF phenotypes. If in HF with reduced ejection fraction metformin had a neutral effect on the risk of death and reduced the risk of hospitalizations, then in HFpEF, metformin therapy was associated with a decrease in the risk of death, as in acute HF. In the abovementioned meta-analysis by Halabi et al., a significantly greater protective effect was observed in patients with EF > 50% (*p* = 0.003).

In patients with diabetes mellitus, metformin improved both the Doppler measurement of the long-axis lengthening rate (e0) and the isovolumic relaxation time, which indicates an improvement in the echocardiography changes that are characteristic of HFpEF [183].

The results of Facila et al. (2016) look impressive: according to this study the use of metformin in the treatment of patients with type 2 DM and acute HF has beneficial effects, reducing the risk of death by 67% (*p* < 0.001), regardless of age, gender, EF, GFR, and prescribed antidiabetic drugs [184]. However, despite growing evidence of the reduced death risk, the current guidelines require discontinuation of metformin therapy in patients with acute conditions associated with the risk of lactic acidosis, such as cardiogenic shock or acute HF [185].

## 7. Metformin in Atherogenesis Inhibition

### 7.1. Experimental Findings Concerning Metformin Effects on Atherogenesis

Currently, a large pool of data has accumulated demonstrating the important effects of metformin on (a) the elastic properties and biological age of the blood vessels, (b) the processes of lipid accumulation in the vascular wall, and (c) the activation processes of macrophages, formation of their various phenotypes, formation of foam cells, and in a clinical aspect, the formation and stability of atherosclerotic plaque. Virtually all signs of early vascular aging (genomic instability, epigenetic changes, telomere attrition, proteostasis, loss of proteostasis, mitochondrial dysfunction, deregulated nutrient sensing, cellular senescence, stem cell exhaustion, altered intercellular communication) can be modulated by metformin [75,186]. These data include the ability of metformin to slow the increase in arterial stiffness and pulse wave velocity and the formation of impaired endothelial function and vasodilation; to inhibit chronic vascular inflammation and intima-media complex thickening; to prevent or to reduce the impairment of the blood rheological properties; to prevent the depletion of the capillary network and its dysfunction; to decrease telomere length and telomerase activity; to prevent impaired glucose and lipid metabolism and oxidative stress; to inhibit arterial calcification, an increase the matrix substances deposition, and disorganization of the small vessels in the kidneys and brain; and to level the increase in load on the LV, reducing its hypertrophy. It is important to note that, even at the IR stage, the ability of adipocytes to accumulate free fatty acids is impaired, and their level in circulation, especially in the postprandial status, increases, which leads to an increase in the de novo synthesis of DAG. This activates PKC, which in turn is responsible for the hyperproduction of endothelial O_2_ radicals and eNOS inhibition. Metformin prevents the adverse effects of excess dietary fat and carbohydrates, including the effects on vascular function [187]. At the DM stage, metformin can reduce hyperglycemia-induced endothelial senescence and apoptosis through a SIRT1-dependent pathway and inhibit OS both under hyperglycemic conditions (in rats and humans) and under fructose or palmitic acid-rich diets. Metformin increases endogenous antioxidant defenses by preventing hyperglycemia-related inhibition of glucose-6-phosphate dehydrogenase, which prevents a decrease in superoxide dismutase-1 production and counteracts the proatherogenic effects of oxidized low-density lipoproteins (oxLDLs) and lectin-like oxLDL receptor LOX-1 [187].

Inhibition of the respiratory complex I in mitochondria by metformin is accompanied by a decrease in ATP production and a concomitant increase in adenosine diphosphate and AMP. This altered cellular energy charge is detected by the main cellular energy sensor, AMPK (Table 2) [116,117]. Experimental studies in ApoE -/- mice have demonstrated the ability of metformin to inhibit atherosclerotic plaque formation, compared to untreated animals (Table 2).

Metformin protects against atherosclerosis by maintaining endothelial integrity and preventing plaque formation by inhibiting lipid entry into the vascular wall. DM is well-known to be characterized by severe plaque instability (Table 2) [142].

In addition, metformin exhibits antithrombotic properties, counteracting the stimulating effect of hyperinsulinemia on the production of an inhibitor of plasminogen activator inhibitor 1 (PAI-1), a negative regulator of fibrinolysis. Metformin directly inhibits the expression of the PAI-1 gene (Table 2). Studies that examined the effect of metformin on the course of acute myocardial ischemia have yielded fewer striking results. In our center, a clinical and experimental study on the effect of metformin on myocardial resistance to ischemia both in diabetes and in its absence was carried out. The effect of metformin on myocardial resistance to ischemic and reperfusion injury was shown only after its intracoronary administration in animals without DM. These results were explained by the highest concentration of the drug achieved in the myocardium. In addition, given that diabetes itself affected ischemic preconditioning of the myocardium, it is possible that the activation of the defense mechanisms in the myocardium associated with the presence of diabetes cannot be enhanced by an additional stimulus [188]. In experimental studies, treatment with metformin was accompanied by the improvement in the functional state of the heart and the inhibition of the HF progression of ischemic and non-ischemic etiology, improving the energy status of the myocardium [189].

### 7.2. Clinical Evidence of the Cardioprotective Effects of Metformin on Atherosclerotic CVD

According to a recent review [187], metformin reduces the risk of the development and the progression of atherosclerosis by reducing the severity of IR and inflammation, most importantly, and a number of CV risk factors (dyslipidemia, hypertension, glycemia, obesity) and by counteracting the proatherogenic role of oxLDL and LOX-1. These effects have been confirmed in studies on patients with and without type 2 DM. We fully support the opinion of the authors that the main atheroprotection effect of metformin is provided by an improvement in insulin sensitivity and, as a result, a decrease in the severity of inflammation and OS.

One of the earliest and longest-running studies demonstrating the positive effects of metformin on the CVD prognosis was the UKPDS study, which demonstrated the ability of metformin to reduce the risk of atherosclerotic cardiovascular events (MI), overall mortality, and diabetes-associated death after the end of the study (UKPDS34) and after 10 years of follow-up (UKPDS80).

The efficacy of metformin in the primary prevention of CV events has been confirmed in a retrospective study [190] that compared 3400 patients with type 2 DM who received metformin and lifestyle modification with patients (*n* = 3400) treated only with lifestyle modification (control group). All patients had no CVD. The average follow-up period was 62.5 months (about 5 years). In the metformin group, there was a decrease in the risk of mortality from all causes by 29.5% (*p* = 0.007), coronary heart disease by 35.5% (*p* = 0.004), chronic HF by 31.2% (*p* = 0.109), and stroke by 30.2% (*p* = 0.024) in comparison with the control group.

Regarding the ability of metformin to provide a secondary prevention of atherothrombotic events, the Reduction of Atherothrombosis for Continued Health (REACH) Registry provided a strong evidence base in the analysis of almost 20,000 patients. A decrease in overall mortality was demonstrated when taking metformin for 2 years (hazard ratio 0.67; 95% confidence interval (CI) 0.59–0.75; adjusted hazard ratio 0.76; 95% confidence interval 0.65–0.89 (*p*-value 0.001 for both, log-rank test)). Hazard ratios were adjusted for age, gender, and significant risk factors [191,192].

Mortality was 6.3% (95% CI 5.2–7.4%) with metformin and 9.8% (95% CI 8.4–11.2%) without metformin (relative risk 0.76 (95% CI 0.65–0.89; *p* < 0.001)). Association with lower mortality was consistent among the subgroups, including those with a history of congestive HF (relative risk 0.69; 95% CI 0.54–0.90; *p* = 0.006), elderly patients (>65 years old (0.77; 0.62–0.95; *p* = 0.02), and patients with GFR 30–60 mL/min/1.73 m^2^ (0.64; 95% CI, 0.48–0.86; *p* = 0.003). Metformin use has been shown to reduce mortality among diabetic patients as a secondary prophylaxis, including those subgroups of patients in which the use of metformin had not been recommended.

Until recently, metformin was not recommended for patients with acute MI [193]; however, publications in recent years have indicated the need to change the current paradigm. They have evidenced a reduced risk of death in patients with MI when receiving metformin, while insulin therapy, on the contrary, worsened the prognosis [194]. Another analysis compared the effect of metformin with sulfonylurea drugs and thiazolidinediones on the outcome of the first MI. Type 2 DM patients receiving metformin during the acute phase of the first MI showed a significantly lower incidence of CV complications (*p* = 0.005) compared with those who did not receive it, including the risk of recurrent MI based on multivariate analysis (hazard ratio 0.33; 95% CI 0.12–0.91; *p* = 0.032). The effect persisted after the adjustment for other risk factors [195].

As noted above, HFrEF is predominantly an outcome of atherosclerotic CVD, especially with acute events (MI), so the effects of metformin on its course should be considered in this section. In the study by Eurich et al., in individuals with chronic HF and low left ventricular ejection fraction, metformin did not increase or decrease the risk of death (relative risk 0.91, 95% CI 0.72–1.14; *p* = 0.34) [196]. At the same time, the use of metformin was associated with a significant reduction in the risk of all hospitalizations by 7% (95% CI 0.89–0.98; *p* = 0.01) without a rise in the risk of lactic acidosis. For many years, metformin has not been recommended for HF because of the risk of increased lactate levels and the development of lactic acidosis. However, the results of clinical observations have shown that this risk is ephemeral, and in reality, metformin therapy improves the survival rate of patients with HF. The mechanism of this effect includes the activation of AMP, which ensures the regression of cardiomyocyte hypertrophy, the suppression of cell apoptosis, the prevention of myocardial fibrosis, and the stimulation of NO synthesis. We will discuss the role of lactate in these processes below. In the later meta-analysis, metformin reduced mortality in HF with both preserved and reduced EF after adjusting for HF treatments such as ACE inhibitors and beta-blockers (β = −0.2 (95% CI −0.3 to −0.1), *p* = 0.02) [197].

## 8. Role of Lactate Elevation in Realization of Metformin Effects

Meanwhile, studies in recent years have changed our view concerning the role of lactate in the organism. An increase in lactate production occurs with a reduction in aerobic glycolysis under hypoxic conditions, which is accompanied by a compensatory increase in the activity of anaerobic glycolysis and the level of lactate in the circulation. In this variant, lactate acts as a marker of metabolic health [198]. Lactate level in the blood in obese patients is significantly higher than in people with normal body weight, reflecting an increase in tissue hypoxia, in particular, hypoxia of adipose tissue, as adipocyte hypertrophy develops. These changes predict the development of oxidative stress and chronic inflammation, which characterize the deterioration of metabolic health. The opposite results were also noted: after bariatric treatment with normalization of the body weight, there was also a decrease in lactate level in the circulation [198]. Moderately elevated lactate levels are often observed in patients with an advanced functional class of HF and in other critical conditions (patients with severe renal failure, brain injury). This, to a certain extent, reflects the characteristics of energy metabolism in hypoxic conditions. New data regarding the role of increased muscle lactate in exercise tolerance have also been obtained. A moderate increase in lactate seems to improve the muscle tolerance to stress, and at high concentrations it worsens. With physical exertion, the need for ATP in the muscles increases with insufficient oxygen supply, and the production of lactate grows. Lactate is involved in muscle fiber interactions (lactate intercellular shuttle) during physical activity. Lactate is actively used by the myocardial fibers and the brain. Thus, in recent years, the place of lactate in energy metabolism has dramatically changed, especially in neural energy metabolism [199]. There is accumulated evidence that lactate may act as a “critical rescue fuel” for the CNS when glucose concentrations fluctuate over a wide range. In the CNS, the lactic acid shuttle of astrocyte neurons provides a mechanism through which astrocytes provide energy metabolism, converting glucose to lactate via glycolysis. This astrocytic lactate spreads from astrocytes to the adjacent neurons, where it is oxidized in the mitochondria (to CO_2_ + H_2_O) to resynthesize ATP and/or be used to produce amino acid neurotransmitters (e.g., glutamate, aspartate, and γ-aminobutyric acid (GABA)). An increased level of lactate in the CNS plays an important role in tumors. Lactate is the “rescue fuel” for the normal CNS cells in cancer, since tumor cells actively exporting lactate have a very low ability to import and use lactate for metabolism. At the same time, tumor cells are more active glucose consumers, which can lead to “consumption hypoglycemia.” Thus, tumor cells actively consume glucose, but they are unable to utilize lactate [200]. Hyperlactatemia is a compensatory mechanism that provides energy to normal cells in cancer, including in conditions of hypoglycemia, and this may be one of the mechanisms for improving the prognosis in cancer patients given metformin.

Today lactate is discussed as an extremely useful energy substrate and anti-inflammatory agent, since in inflammatory processes it can inhibit inflammasomes in traumatic brain injury; in acute damage of the pancreas and the liver; in MI, heart surgery and acute HF; and in several other urgent situations [201]. At the same time, the regulation of redox/ROS, calcium/calmodulin-activated protein kinase II (CaMK II)/PKC and PGC1a has been discussed as the key mechanisms of its action, which indicates the role of lactate in the realization of metformin’s effects. Therefore, the view of metformin therapy as an option that must be interrupted in urgent situations because of the risk of lactic acidosis may be outdated. The review discussing the risk of lactic acidosis from metformin therapy claimed that metformin-induced lactic acidosis arises only in cases where three conditions are met: lactate level >5mmol/L, pH < 7.35, and metformin concentration in the circulation >5mg/L. Only 10% of cases of lactic acidosis described as metformin-associated met these criteria [202]. In most cases that are recorded as metformin-induced LA, the latter only limited the ability of patients to cope with an increase in lactate levels caused by another event (not metformin) that triggered LA. Moreover, LA patients treated with metformin have a significantly better prognosis than LA patients not receiving metformin [203]. Thus, the risk of lactic acidosis developing from metformin is seriously exaggerated, and in most cases, this therapy gives only moderate hyperlactatemia, which does not worsen the prognosis in urgent situations, but, on the contrary, can improve the energy metabolism and survival. In addition, the latest theories of diabetes development have put redox stress on the list of the earliest changes, appearing long before the development of carbohydrate metabolism disorders inducing a cascade of subsequent pathological events. This once again leads us to the usefulness of the earliest administration of metformin in situations with a high risk of prediabetes and diabetes development.

It is possible to draw an analogy to the effects of SGLT2 inhibitors. The increase in the production of ketones allows these drugs to provide a convenient and economical treatment in terms of an oxygen consumption variant of the energy substrate, which is much more suitable for organs in the state of hypoxia. Energy metabolism changes from lipolysis to ketolysis in damaged kidneys [204] and heart [205]. Hyperactivation of mTORC1 leads to a decrease in renal lipolysis with subsequent renal damage. Elevated ketone bodies increase the severity of renal damage by blocking mTORC1 signals. Thus, the mechanisms of reno- and cardioprotection of SGLT2 inhibitors include the inhibition of mTORC1 by ketone bodies [204], and this largely resembles the effects of metformin, also including the inhibition of mTOR and providing an energy substrate suitable for use under hypoxic conditions (lactate).

## 9. Conclusions

Summing up, we note that the accumulated data indicate the possibility of expanding the boundaries of metformin use toward more new targets: anti-aging therapy [186], improvement in the treatment of many types of cancer [205], renal pathology of both diabetic and non-diabetic (toxic and ischemic) genesis, many profibrogenic diseases, and in the direction of expanding the possibilities of its use in diabetes mellitus and in risk groups for its development. At the same time, we are moving toward prescribing it increasingly earlier in treatment, and it can be assumed that new large studies will prove that the moment to start metformin should occur earlier than prediabetes in situations with a high risk of developing metabolic disease. These include an unhealthy lifestyle with intake of high animal fat and high glycemic index food, low physical activity, increasing excess weight with a tendency to visceral fat deposition. Recent studies have shown that already at this stage an imbalance in microbiota phenotypes has developed with a decrease in the number and diversity of butyrate-producing and lactate-producing bacteria, impaired production of incretins, and the formation of redox stress and oxidative stress. This will lead to a future deterioration in secretion and a decrease in the number of β-cells, development of hyperglucagonemia, and progression of metabolic and vascular changes whose reversibility is debatable. Intervention with metformin as early as possible protects one, to a certain extent, from the negative influences of an unhealthy lifestyle, giving the patient a temporary head start to change it. On the other hand, we have received more and more data about conditions that were previously considered as contradictions to metformin use where metformin therapy improved the prognosis compared to most antidiabetic drugs. In these cases, we consider the ability of metformin to increase resistance to hypoxia and cell survival under hypoxic and toxic conditions as a key mechanism. Moderate hyperlactatemia accompanying metformin therapy may play a significant role in these adaptive mechanisms. Further studies are needed to finally resolve the question of lifting restrictions on the use of metformin in acute heart failure, in the acute period of MI, and in a number of other acute conditions, including infectious diseases (COVID-19), considering the recent data showing a significant improvement in their prognosis with metformin therapy.

## Figures and Tables

**Table 1 ijms-23-02363-t001:** Key effects of metformin at different stages of disorders of carbohydrate metabolism.

Stage	Key Pathogenetic Disturbances	Key Effects of Metformin	References
No obesity, no carbohydrate metabolism disorders, leading an unhealthy lifestyle (consuming hypercaloric, fat-rich and high glycemic index food) (results of clinical and experimental studies)	high-fat and carbohydrate-rich food with a high glycemic index → change of microbiome composition: - ↓ abundance of Bacteroidetes → ↓ production of SCFAs (acetate and propionate) which improve insulin sensitivity; - ↓ abundance of bacteria of the genus Lactobacillus → ↓ production of their metabolites (lactate), which are key modulators of glucose metabolism in the gastrointestinal tract and expression of SGLT-1; - ↓ abundance of butyrate-producing bacteria (*Butyricimonas* spp. and *Allobaculum*) → ↓ production of their metabolite (butyrate), which increases insulin sensitivity and the secretion of intestinal hormones, in particular incretins	despite maintaining a high-fat diet, restoration of the abundance of: - Bacteroides, the genus of the phylum Bacteroidetes; - butyrate-producing bacteria in the intestine (*Butyricimonas* spp. and *Allobaculum*); - Parabacteroides, a succinate producer ↑ concentration of SCFAs (butyrate and propionate) in fecal samples from people receiving metformin	[19,20,21,22,23,24,25,26,27,28,29,30]
	↓ secretion of incretins (mainly GLP-1) and sensitivity to them (GLP-1 and GIP)	improving incretin secretion and sensitivity to them: stimulation of SGLT-1 → ↑ GLP-1 release in the experiment: ↑ production of GLP-1, but not GIP; activation of the expression of tissue receptors of GLP-1 and GIP, ↑ tissue sensitivity to both incretins	[31]
	high-fat diet → ↑ intestinal permeability for LPS, ↓ abundance of A. muciniphila, ↓ mucin production, ↓ its anti-inflammatory effects with ↑ levels of IL-6 and IL-1β	modulation of the expression of the MUC2 and MUC5 genes → ↑ mucin level, ↑ abundance of A. muciniphila, which is involved in mucin production → ↓ intestinal permeability for LPS ↑ MUC2 expression → ↑ production of mucin proteins zonulin-1 and occludin → ↓ intestinal permeability ↑ abundance of A. muciniphila → ↓ level of IL-6 and IL-1β	[19,20,22,32,33,34,35]
	↓ production of secondary bile acids (deoxycholic acid and lithocholic acid), that are formed with the participation of enzymes and intestinal microbiota and play a significant role in glucose and lipids metabolism in the gastrointestinal tract	slowing down bile acids metabolism in the intestine → prolongation of bile acids action → improving of the metabolism of lipids and glucose	[36,37,38,39]
	modulation of intestine–CNS axis: high-fat diet → impairment of the production of microbiota metabolites (SCFAs), which have a multilevel effect on the regulation of eating behavior SCFAs → ↑ secretion of intestinal hormones (serotonin, ghrelin, CCK, PYY and GLP-1), which regulate the secretion of insulin, gastric juice and bile acids, and VN activity A number of SCFAs (acetate, butyrate) → penetration through BBB → direct participation in the regulation of satiety and inhibition of inflammation of CNS SCFA → inhibition of fat accumulation in the adipose tissue, enhancing its utilization, and improvement of sensitivity to leptin and ghrelin	restoration of the abundance of bacteria of the genus Lactobacillus and ↑ abundance of Bacteroides, the genus in the phylum Bacteroidetes, butyrate-producing bacteria (*Butyricimonas* spp. and *Allobaculum*), Parabacteroides, producing succinate under the influence of metformin, despite maintaining a high-fat diet ↑ production of SCFAs, evidenced by an ↑ concentration of SCFA (butyrate and propionate) in fecal samples from people receiving metformin improvement in the functional state of the intestine–CNS axis	[36,37,38,39]
Obesity	↓ Bacteroidetes in relation to Firmicutes change in the contribution of Actinobacteria, ↓ in the number of butyrate and lactate-producing bacteria	restoration of the abundance of bacteria of the genus Lactobacillus and ↑ in the abundance of Bacteroides, the genus in the phylum Bacteroidetes, butyrate-producing bacteria (*Butyricimonas* spp. and *Allobaculum*)	[34,35]
	↑ plasma levels of LPS after high-fat meals compared with people without obesity ↑ penetration of LPS into the bloodstream, their accumulation in adipocytes with the development of their hypertrophy, insulin resistance, inflammation, internalization of LPS-lipoproteins by adipose tissue macrophages, with a change in their phenotype from M2 to M1—the development of hypertrophic, metabolically unhealthy obesity	protective effect in LPS-induced damage to epithelial cells of the respiratory tract in the experiment: anti-inflammatory effects, induction of ATF-3 in parallel with protective effects against lipoprotein-induced inflammation in the experiment: suppression of LPS-induced response of macrophages and resolution of allergic dermatitis by modulating autophagy ↓ production of pro-inflammatory cytokines in LPS-stimulated cells	[40,41,42,43]
Type 2 diabetes mellitus	greater ↓ Bacteroidetes and ↑ pool of Firmicutes Proteobacteria, ↓ bacteria of the genus Roseburia, a butyrate producer abundance of *Lactobacillus* spp. was higher, than in healthy people which is regarded as an attempt at immunomodulation ↑ abundance of Gram-negative bacteria → stimulation of the immune system through TLR and development of insulin resistance	suppression of TLR 4 signaling, including after myocardial infarction, weakening left ventricle dysfunction; in myocardial dysfunction caused by sepsis; in lung endotheliocytes	[44,45,46,47,48,49,50,51,52,53]
Latent and transient stage of diabetes	islet redox stress (oxidative and nitrosative stress), free-radical polymerization of islet amyloid polypeptide monomer, β-cell stress of endoplasmic reticulum, UPR stress at transient stage: joining of the processes of amyloidosis and fibrosis of β-cells → development of carbohydrate metabolism disorders	regulating the activity of GRP78 → reduction of redox stress and normalization of the formation of the correct configuration of proteins	[54,55,56,57,58]

SCFAs—short-chain fatty acids; GLP-1—glucagon-like peptide-1; GIP—glucose-dependent insulinotropic peptide; SGLT-1—sodium glucose cotransporter-1; LPS—lipopolysaccharides; IL-6—interleukin-6; IL-1β—interleukin 1β; VN—vagus nerve; CCK—cholecystokinin; PYY—peptide YY; CNS—central nervous system; UPR—unfolded protein response; GRP-78—glucose regulatory protein 78; TTLR—Toll-like receptors; DM—diabetes mellitus; ATF-3—transcriptional activation factor 3; →—affect; ↑—increased; ↓—decreased.

**Table 2 ijms-23-02363-t002:** Key effects of metformin at different stages of the cardiometabolic and cardiorenal continuum.

Stage	Key Pathogenetic Disturbances	Key Effects of Metformin	References
Development of HFpEF and CKD in patients with metabolically unhealthy obesity, prediabetes, and early diabetes	excess nutrients → functional overload of mitochondria → violation of autophagy processes → ↑ ROS → toxic effects on cell structures and ↓ SIRT1/PGC-1a/FGF21 and ↓ AMPK	activation of AMPK → AMPK-mediated inactivation of mTOR → ↑ mitochondrial biogenesis and aerobic glycolysis → improvement of autophagy processes → ↑ collagen turnover through autophagy	[88]
	hyperinsulinemia and IR → ↑ activity of NHE1 in the heart and NHE3 activity in the kidney → ↑ circulating blood volume and sodium retention → ↑ LV filling pressure	↓circulating insulin levels and improvement of insulin sensitivity through multiple mechanisms considering that hyperinsulinemia is a key factor in increasing the activity of NHE1 and NHE3, it can have at least indirect effects on these mechanisms	[89,90]
	↑ volume and change in characteristics of epicardial adipose tissue (a change in the phenotype of adipocytes from brown to white), ↑ production of pro-inflammatory adipokines, the development of inflammation and myocardial fibrosis	↓production of pro-inflammatory cytokines and anti-fibrotic effect	[91,92,93,94]
	metabolic unhealthy obesity→ hyperinsulinemia and lipotoxicity → activation of systemic inflammation and formation of AGEs → ↓ NO synthesis and ↑ROS production→ induction of oxidative stress and the accumulation of peroxidation products, ↓ activity of PKG in cardiomyocytes → LVH oxidative stress activation → mitochondrial dysfunction → ↓ ATP production, ↓ calcium release from the sarcoplasmic reticulum, ↓ SERCA activity and ↓ sensitivity of myofibrils to calcium, ↓ activity of GTP4 violation of the AGE/RAGE ratio → ↑ TGF-β production → induction of fibrosis	inhibition of TGF-β production → ↓ phosphorylation and nuclear translocation of Smad2/3 and preventing the transcriptional activation of fibrogenic target genes such as collagen 1α1 (col1a1) and collagen 3α1 (col3a1) modification of integrin expression → ↓ phosphorylation of ERK1/2 and improving the expression of extra cellular matrix components, inhibition of the expression of TGF-β1-induced monocytic chemotactic protein-1, reduction in the activity of p38 MAPK and JNK, block of the effect of TGF-β1 on phosphorylation of GSK-3β and nuclear translocation of β-catenin → inhibiting the transcription of various fibrogenic genes, including fibronectin acceleration of the fibrosis resolution: - reduction in the level of Gal-3 in the systemic circulation, in adipocytes and monocytes in patients with type 2 DM - ↓ NOX4 activity, mitochondrial oxidative stress and inhibited activity of PKCα and mTOR-S6K signaling pathway → ↓ expression of Gal-3 secreted by cardiomyocytes in mice, ↓ Gal-3 in bloodstream, adipocytes, monocytes → ↓ activation of cardiac fibroblasts → ↓myocardial fibrosis → improvement in cardiac fibrosis after myocardial infarction in mice	[95,96,97,98,99,100,101,102,103,104,105,106,107]
	hyperuricemia → inhibition of AMPK activity → induction of oxidative stress hyperuricemia → IRS/PI3K/Akt pathway → induction of IR in cardiomyocytes, adipocytes, muscles and liver hyperuricemia → inhibition of insulin signaling and induction of IR of cardiomyocytes in vitro and in vivo hyperuricemia → reduction in the speed of GLUT4 movement → inhibition of insulin-induced glucose uptake in cardiomyocytes → ↑ ROS production hyperuricemia is characterized by ↑inflammatory markers (CRP), fibrosis markers —Gal-3 and CITP	activation of AMPK and its phosphorylation → reduction in the severity of hyperuricemia and blocking of its negative effects → protection against hyperuricemia-induced IR in cardiomyocytes and skeletal muscles and associated pathological processes, including fibrogenesis (↑ levels of Gal-3, types 1 and 3 procollagen)	[108,109,110,111,112]
Development and progression of atherosclerosis in type 2 DM	changes in the activity of a number of genes and transcription factors, for example, NF-κB, and molecules, such as AGE, capable of modifying components of the extracellular matrix → ED activation of the RAGE in vasculature in DM → atherogenesis hyperglycemia → transcription factor NF-κB → development of ED, modulation of the expression of a number of microRNAs (in particular, miR-126, -21, and miR-146a-5p) involved in atherogenesis, and PKC hyperactivation → ↑ production of superoxide anions and VEGF, ↓ NO production, ↑ activity of the polyol pathway → consumption of NADPH enhances intracellular oxidative stress hyperglycemia → impairment of vascular permeability and involvement of leukocytes in inflammatory reactions → changes in EC morphology and density, changes in the functional properties of EC → deterioration in the bioavailability of NO, modulation of vascular tone, a violation of the ratio of vasodilators (NO and PGI2) and vasoconstrictors (ET-1)	activation of AMPK → delay of endothelial and vascular aging, ↑rate of oxygen consumption by mitochondria AMPK-dependent ↑ H3K79me3 → SIRT1-DOT1L → hTERT (enzyme involved in adding telomere repeats to the ends of the chromosome, a process that modulates vascular senescence) → reduction of stiffness of the vascular wall and deceleration of vascular aging AMPK-dependent H3K79me → ↑ SIRT3 → improvement of mitochondrial biogenetics/function and delay of endothelial aging in patients with prediabetes: ↑ SIRT1 expression, ↓ p70S6K phosphorylation and improvement of plasma N-glycan profile → ↑ telomere length in mononuclear cells, which, as previously shown in the experiment, regulate lifespan	[113,114,115,116,117]
	hyperglycemia → disruption in the functioning of the insulin receptor, activation of transduction in favor of proatherogenic effects instead of antiatherogenic ones → activation of atherogenesis insulin receptor → activation of PI3K/protein kinase B (Akt)/eNOS pathway in the endothelium → phosphorylation of eNOS at Ser1177 →activation of phosphorylation of SHC, activating MAPK pathway → ↑ ET-1 expression, ↓ NO availability → impairment of NO functions: maintaining vascular homeostasis, protecting against the development of ischemic heart disease, ↑ mitochondrial organization by ↑AMPK activation and PGC-1a expression hyperglycemia → deficiency of proteoglycans, which inhibit the binding of monocytes to the subendothelial region. oxLDLs → activation of macrophages → rapid progression of atherogenesis	AMPK → inhibition of SREBP-1 and ChREBP → reduction in the expression of several genes of lipogenesis inhibiting the entry of lipids into the vascular wall → prevention of plaque formation AMPK → phosphorylation of eNOS at Ser 1179 → ↑ bioavailability of NO experimental studies: affecting to hematopoietic AMPK → direct inhibition of atherogenesis	[118,119]
	hyperglycemia → oxidative stress, oxidation of LDL, ↑ activity of the hexosamine pathway → fructose-6-phosphate, instead of being included in glycolysis, becomes a substrate for GFAT → TGF-β transcription and fibrotic processes	inhibition of glycerophosphate transporter enzyme mGPDH → prevention of the use of glycerol as a substrate for gluconeogenesis	[120]
	↑ AGEs and other glycation products in EC → inhibition of the selective uptake of HDL ester and the efflux of cholesterol from peripheral cells to HDL HDL glycation → HDL dysfunction → loss of their atheroprotective effects	AMPK activation and RAGE/NFκB pathway suppression → inhibition of AGE products-induced inflammatory response in murine macrophages	[120,121]
	high levels of glucose, modified lipoprotein particles and saturated fatty acid particles →↑ inflammation in vascular EC: NF-κB → activation of inflammatory mediators (TNF-α, IL-1β, IL-6), PKC, CAM →facilitation of the adhesion of monocytes and T-cells to EC	reduction of production of pro-inflammatory cytokines, primarily TNF-α, IL-1β, IL-6	[122]
	changes in miRNA expression (miR-126, miR-21, miR-146a-5p) activation of miR-21: - activation of endothelium with the formation of foam cells, adhesion of circulating monocytes, apoptosis of macrophages and the ability of phagocytes to clearance - suppression of the main mitochondrial antioxidant enzyme SOD2 → inhibition of the antioxidant response, and promotion of proinflammatory reactions in EC - induced by hyperglicemia binding of NF-κB to the miR-21 promoter→ initiation of its transcription ↓expression of miR-146a-5p: - weakening of inhibition of target genes involved in NF-κB and other pathways of cytokine production and signaling - impairment of its effects on NF-κB signaling - negative modulation of IL-6 → growth and destabilization of atherosclerotic plaques	significant change in expression profiles of a number of miRNAs, including -miR-21-5p, miR-126-5p, miR-146a-5p reduction of of miR-21-5p expression in CVD reduction in miR-126-5p expression in type 2 DM ↑146a-5p expression in obesity, CVD	[123,124,125,126,127,128,129,130,131,132,133,134]
	imbalance of the AGE/ RAGE axis, ↑ glycosylation of O-GlcNac and stimulation of osteogenesis regulator Runx2, hyperglycemia-induced ↑ levels of TGF-β1 and osteogenic markers (alkaline phosphatase, osteocalcin, Runx2) → deposition of crystals of hydroxyapatite (calcium or phosphate) and elastin degradation products in vascular cells → activation of vascular calcification → plaque rupture, cardiovascular events	AMPKα1-dependent pathway → reduction in atherosclerotic calcification and Runx2 expression in ApoE mice AMPK/eNOS/NO signaling pathway → block of vascular calcification inhibition of the PDK4/oxidative stress-mediated apoptotic pathway through enhanced mitochondrial biogenesis → attenuating β-GP-induced conversion of the vascular SMC to the osteogenic phenotype	[135,136,137,138,139,140]
	ATF1 → determination of macrophage Mhem phenotype → intra-plaque hemorrhages → instability of atherosclerotic plaques	AMPK → inhibition of Mhem macrophages and foam cell formation	[88,141,142]

AMP—adenosine monophosphate; AMPK—AMP-activated protein kinase; ROS—Reactive oxygen species; IR—insulin resistance; HFpEF—heart failure with preserved ejection fraction; CKD—chronic kidney disease; SIRT1—sirtuin 1; NHE 1—sodium-hydrogen exchanger type 1; NHE 3—sodium–hydrogen exchanger type 3; AGEs—advanced glycation end products; NO—nitric oxide; PKG—protein kinase G; LVH—left ventricle hypertrophy; SERCA—sarcoendoplasmic reticulum calcium ATPase; ATP—adenosine triphosphate; RAGE—receptor for advanced glycation end products; TGF-β—transforming growth factor β; GTP-4—glucose transport protein 4; ERK1/2—extracellular signal-regulated kinase ½; JNK—c-Jun N-terminal kinase; GSK-3β—glycogen synthase kinase 3β; MAPK—mitogen-activated protein kinase; NOX4—reduced nicotinamide adenine dinucleotide phosphate (NADPH) oxidase 4; PKCα—α protein kinase C; Gal-3—galectin 3; GLUT4—glucose transporter type 4; CRP—C-reactive protein; CITP—carboxy-terminal telopeptide of type I collagen; NF-κB—nuclear factor κB; ED—endothelial dysfunction; VEGF—vascular endothelial growth factor; miR—microRNA; EC—endothelial cells; PGI2—prostacyclin; ET-1—endothelin-1; DOT1L—disruptor of telomeric silencing-1 like protein; hTERT—human telomerase reverse transcriptase; PI3K—phosphatidylinositol 3-kinase; eNOS—endothelial nitric oxide synthase; SHC—Src-homology 2 domain contain; OxLDLs—oxidized low-density lipoproteins; SREBP-1—sterol regulatory element binding protein; ChREBP—carbohydrate response element binding protein; GFAT—glutamine fructose-6-phosphate amidotransferase; mGPDH—mitochondrial glycerophosphate dehydrogenase; TNF-α—tumor necrosis factor α; IL—interleukin; SOD2—superoxide dismutase 2; PKC—protein kinase C; CAM—cell adhesion molecules; CVD—cardiovascular diseases; O-GlcNac—N-acetylglucosamine; Runx2—Runt-related transcription factor 2; PDK4—pyruvate dehydrogenase kinase; β-GP—β-glycerophosphate; SMC—smooth muscle cells; →—affect; ↑—increased; ↓—decreased.

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
