# Peer review of "Metformin: Expanding the Scope of Application—Starting Earlier than Yesterday, Canceling Later"

_ijms, 2022, doi:10.3390/ijms23042363_

Round 1
Reviewer 1 Report
This is a well written, interesting review article that will contribute to the literature.
Author Response
Dear reviewer! We are grateful for your review. Thank you for a high estimation of our manuscript.
Reviewer 2 Report
The manuscript is interesting and certainly of interest to readers.
This review requires to enrich the manuscript with some very updated references.
1- In this review the authors discuss the removal of a number of restrictions on metformin use and they present a new perspective on the role of increasing lactate in metformin therapy. In this context, a recent review on the risk of lactic acidosis has offered an important and updated overview on this issue (Diabetes Res Clin Pract. 2019 Nov;157:107879. doi: 10.1016/j.diabres.2019.107879.). This interesting reference should be discussed in the text.
2- A large space in the manuscript it is dedicated to the possible advantage of metformin towards HF. A very recent paper collects the main pathophysiological and clinical evidence of this use of metformin (Biomolecules. 2021 Dec 4;11(12):1834. doi: 10.3390/biom11121834.). The above reference should be added in the text.
3- The beneficial effects of metformin on endothelial dysfunction in DM2, which could prevent the development of atherosclerosis, were detailed in a recent article (Biomedicines. 2020 Dec 22;9(1):3. doi: 10.3390/biomedicines9010003.). It would be appropriate to comment and add the above reference in the manuscript.
4- The authors should discuss the rationale for the use of metformin as part of a combination therapy in a variety of clinical settings, for example allowing for a reduction in the dose of chemotherapy in cancer patients. In particular, the action of metformin on the RAS/RAF/MAPK pathway should be focused, as well as on the anti-aging action of metformin (ESMO Open. 2017 May 2;2(2):e000132. doi: 10.1136/esmoopen-2016-000132. - Diabetes Res Clin Pract. 2020 Feb;160:108025. doi: 10.1016/j.diabres.2020.108025.). These issues and above references should be incorporated and discussed in the text.
5- Finally, tables 1 and 2 should be better formatted to be more readable.
Author Response
Dear reviewer! We are grateful for your review and for valuable comments.
We have carefully reviewed your comments and have revised the manuscript. Changes to the manuscript according to your comments are shown in blue.

Reviewer 3 Report
The present paper summarizes a number of studies involving the use of metformin under different circumstances and for different types of therapeutical effect, as well as the conclusions of those studies. The authors have put a lot of effort in the review of all these scientific publication and have done an admirable job summarizing the data and drawing their own conclusions.
However, for the most part the manuscript needs extensive editing, English grammar needs major improvements, as well as the presentation of the manuscript needs to be made much more homogenous. Besides the numerous grammatical errors, there are a number of instances where font type and size vary greatly within the same table, as does the writing style, letter capitalization and punctuation (this is true for all tables).
What is the reason for which, as stated in the conclusions, in the future there is a "high likeliness" that "restrictions in metformin use" "will be removed"? Please elaborate more on the objective considerations based on which you draw the conclusions of the manuscript, taking into account ethical and legal implications. Or, alternatively, I would suggest a more weighted approach to formulating conclusions.
Author Response
Dear reviewer!
We express our deepest gratitude to you for your work devoted to the analysis of our manuscript, for your extremely valuable comments.
Based on your recommendation, we have sent the article for language editing (certificate is attached). We have tried to improve the structure of the article and have softened the wording in the conclusion. Tables 1 and 2 have been completely revised and structured.
We sincerely hope that we have managed to correct our shortcomings. Corrections made in accordance with your recommendations are highlighted in red in the text.

Round 2
Reviewer 3 Report
Dear authors,
I would like to sincerely congratulate you for the extensive re-editing of the manuscript. The style is much more homogenous, data better organized and English grammar and spelling dramatically improved.
The conclusions were also balanced better in tone and scientific soundness.
As is, I will recommend publication of your manuscript and would once again like to congratulate you on your hard work.